

# Changes in intentional binding effect during a novel perceptual-motor task

Shu Morioka[1], Kazuki Hayashida[2], Yuki Nishi[2], Sayaka Negi[3], Yuki Nishi[4], Michihiro Osumi[1] and Satoshi Nobusako[1]

[1] Neurorehabilitation Research Center, Kio University, Kitakatsuragi, Nara, Japan
[2] Graduate School of Health Sciences, Kio University, Kitakatsuragi, Nara, Japan
[3] Department of Rehabilitation, Kishiwada Rehabilitation Hospital, Kishiwada, Osaka, Japan
[4] Department of Rehabilitation, Ishida Clinic, Osaka, Japan

Corresponding author
Shu Morioka, s.morioka@kio.ac.jp

## ABSTRACT

Perceptual-motor learning describes the process of improving the smoothness and accuracy of movements. Intentional binding (IB) is a phenomenon whereby the length of time between performing a voluntary action and the production of a sensory outcome during perceptual-motor control is perceived as being shorter than the reality. How IB may change over the course of perceptual-motor learning, however, has not been explicitly investigated. Here, we developed a set of IB tasks during perceptual-motor learning. Participants were instructed to stop a circular moving object by key press when it reached the center of a target circle on the display screen. The distance between the center of the target circle and the center of the moving object was measured, and the error was used to approximate the perceptual-motor performance index. This task also included an additional exercise that was unrelated to the perceptual-motor task: after pressing the key, a sound was presented after a randomly chosen delay of 200, 500, or 700 ms and the participant had to estimate the delay interval. The difference between the estimated and actual delay was used as the IB value. A cluster analysis was then performed using the error values from the first and last task to group the participants based on their perceptual-motor performance. Participants showing a very small change in error value, and thus demonstrating a small effect of perceptual-motor learning, were classified into cluster 1. Those who exhibited a large decrease in error value from the first to the last set, and thus demonstrated a strong improvement in perceptual-motor performance, were classified into cluster 2. Those who exhibited perceptual-motor learning also showed improvements in the IB value. Our data suggest that IB is elevated when perceptual-motor learning occurs.

## INTRODUCTION

Humans combine their motor intentions with the sensory feedback from the completed action to modulate behavior when experiencing an error (i.e., when the sensory output is not in line with the original intention) (*Wolpert, 1997*; *Imamizu et al., 2000*). Such refinement of behavior based on error feedback is known as learning. Modulating behavior during a voluntary action is achieved by comparing the internal prediction of the outcome

with the outcome of the motor action (a mechanism of perceptual-motor learning). Perceptual-motor learning also includes an aspect of reinforcement learning, where the behavioral goal is rewarding and as such, the motor action is repeated to improve precision (*Doya, 2000*; *Schultz, 2006*).

Intentional movements that are made with a specific goal or purpose are governed by a smooth and flexible feedback mechanism that coordinates motor prediction, motor command, and afferent sensory feedback. These three processes are interconnected via an internal, forward moving model, referred to as the internal predictive model (*Blakemore, Wolpert & Frith, 2002*; *Wolpert & Ghahramani, 2000*). The cognitive mechanisms underlying these comparisons are called comparator models and form the neural basis of motor control (*Wolpert, 1997*). Thus, the comparator model was originally developed to explain how the brain monitors intentional movements. A similar model has also been used to explain sense of agency (SoA) (*Haggard, 2017*). SoA is a conscious experience of viewing one's behavior: an individual's intentions are the cause of a specific event in the external world. In other words, SoA is the sensation that "*I am the one who is causing or generating an action*" (*Gallagher, 2000*). Thus, it has been hypothesized that similar models can be adopted for both motor control or learning and SoA. This hypothesis, however, remains to be demonstrated experimentally. Although similar mechanisms may underlie both perceptual-motor learning through error correction and SoA generation, previous studies have only examined these processes independently. According to a previous survey, the comparator model suggests that SoA arises from the comparison between the predicted and actual sensory feedback (*Frith, Blakemore & Wolpert, 2000*). If the predicted sensory effect matches the actual sensory effect, a sensation is perceived as self-caused. The comparator model, however, cannot explain some aspects of the experience of agency. For example, not all divergences from the predicted sensory effect reach awareness, and small sensory discrepancies or their ensuing motor adjustments do not necessarily influence the SoA (*Castiello, Paulignan & Jeannerod, 1991*; *Fourneret & Jeannerod, 1998*). Voluntary actions are also important for generating SoA (*Haggard, Clark & Kalogeras, 2002*), and SoA increases when there is a strong motivation for the behavior or a clear behavioral objective (*Bandura, 2001*). In other words, SoA may be reinforced in an ambitious motor learning task with a goal.

The concept of intentional binding (IB) has gained recent attention as a factor related to SoA (*Haggard, 2017*). Binding effects are often measured by Libet's clock method. Here, participants press a key at a time of their choosing, which results in a tone being produced after a 250 ms delay. The participants must then judge where the clock hand was when they pressed the key or when they heard the tone, in separate blocks of trials (*Haggard, Clark & Kalogeras, 2002*). Another method to measure binding effects involves estimating the interval between pressing a key and hearing an auditory stimulus without using a clock, thus directly measuring the shift of the action and its consequence (*Moore, Wegner & Haggard, 2009b*). The perceived time interval is used as an index of the binding effect, whereby shorter intervals indicate greater binding effects. Changes in IB during perceptual-motor learning, however, were not investigated in the previous studies that used a traditional IB task. IB is a phenomenon whereby the length of time

between a voluntary action and the production of a sensory outcome is perceived as being shorter than the reality (*Haggard, Clark & Kalogeras, 2002*). An IB effect is not observed for involuntary actions, implying that IB is an index for SoA. Because the amount of shift in duration is caused by a voluntary movement, it is considered to reflect SoA. This phenomenon is often considered to be due to experiencing agency (*Haggard, 2005*; *Tsakiris & Haggard, 2005*), as the temporal compression of the interval between actions and consequences may help an individual determine whether they were responsible or not for a sensory event. IB is thought to result from forward action models, such that when an action effect is predicted it results in an altered experience of the sensory event. One study reported that the IB effect is enhanced during the act of key pressing when the action intention is formed in advance, compared to when it is simply an action of key pressing (*Vinding, Pedersen & Overgaard, 2013*). Therefore, we can speculate that IB is elevated when the action intention is formed during a perceptual-motor learning task with a goal.

Previous research suggests that the shift of the perceived time of an action and its outcome is caused by postdictive and predictive processes, respectively (*Moore et al., 2010b*). In addition, several studies have shown that predictive and postdictive mechanisms are responsible for the emergence of IB (*Moore et al., 2009a*; *Moore, Wegner & Haggard, 2009b*). *Moore et al. (2009a)* manipulated the probability of occurrence of the result (tone) and experimentally proved an element of the postdictive process that the higher the probability of occurrence, the greater the binding effect. In addition, *Moore & Haggard (2008)* demonstrated that binding occurred even if no result (tone) was obtained in cases where the predictability of the effect of the action was high. They thus concluded that predictive processes affect binding.

Others have used an experimental paradigm where the actions of the participant results in an audible tone that is associated with monetary rewards or penalties. In one such study, IB was reduced in the penalty trials compared to the neutral or reward trials (*Takahata et al., 2012*). These findings indicate the impact of rewards and penalties on the effects of IB. Dopaminergic neurons function in the reward system, and are heavily involved in reinforcement learning. Interestingly, dopaminergic medication, such as Levodopa, increases the binding effect in patients with Parkinson's disease—a condition characterized by dopamine depletion (*Moore et al., 2010a*). Dopaminergic activation can't be elicited by tasks that are too difficult or too easy.

Other models have emphasized the effect of IB by retrospective processes that arise after the occurrence of the action outcome (*Maeda et al., 2012*; *Wegner, 2003*). However, temporal changes in binding effects during a perceptual-motor learning process in terms of postdictive mechanisms have not been studied. A recent study by *Di Costa et al. (2018)* investigated binding effects within a dynamic environment for reinforcement learning in which participants were encouraged to achieve goals. The researchers found that (1) negative outcomes modestly increased the binding effect, (2) errors were important for adjusting and executing the next action, and (3) that SoA was enhanced post error. This study revealed a correlation between the post-error

binding effect and the learning effect. However, the time-series variation of the binding effect was not investigated, and the level of difficulty in the learning task and individual differences in learning ability were not considered.

Based on the above discussions, it is important for a perceptual-motor task to have an optimal difficulty level. Our current study aimed to elucidate how the binding effect is modulated with learning progression. To address this aim, we developed an IB task that incorporated a perceptual-motor learning task into the experimental task paradigm used in previous IB-related studies. A previous study tested the relationship between motor control and IB by administering noise and changing the motor control of study subjects (*Kumar & Srinivasan, 2017*). This approach, however, did not report the learning ability of the subjects themselves. Instead, our study focused on testing the relationships between the transition of individual learning ability and IB. The subjects included in our study were classified by clustering to consider individual learning ability for the learning task, that is, the level of difficulty. This strategy allowed us to clarify whether errors during learning contribute to the increase in IB based on time-series data. We hypothesized that an IB effect would be observed when a perceptual motor learning proceeds (i.e., a decrease in the error frequency occurs), but that once learning stagnates, this IB effect would be smaller. We considered that the learning process could be clarified by arranging the traditional IB task. We found that errors arising during a perceptual-motor task may help enhance binding effects.

## METHODS

### Participants
A total of 30 healthy university students (14 men and 16 women; age: Mean, 21.50 years; standard deviation, 0.56 years) were recruited from Kio University for this study. The experimental procedure was outlined to the participants prior to the experiment but the purpose of the experiment was not explained in order to prevent participant bias. The experimental protocol was approved by the University Ethics Committee (H28-50). All participants provided written informed consent prior to their participation in the experiment.

### Devices and software
LabVIEW (National Instruments, Minato-ku, Japan) was used to design the IB task paradigm, and a 23-inch display (1,098 × 630 px) was used as the presentation screen (Mitsubishi Electric, Tokyo, Japan). A personal computer (Twotop Original PC, Unitcom, Japan) was used for all tasks and to record the data. The refresh rate of the display was 60 Hz, and the size of the stimulus in visual angle (deg) was 43.6° horizontally, 21.8° vertically, and the viewing distance was 75 cm.

### Procedure
We developed a task to extract indices of IB and perceptual-motor performance based on the interval estimation paradigm (*Engbert et al., 2007*) and with the goal of investigating the relationship between IB and perceptual-motor performance (Fig. 1).

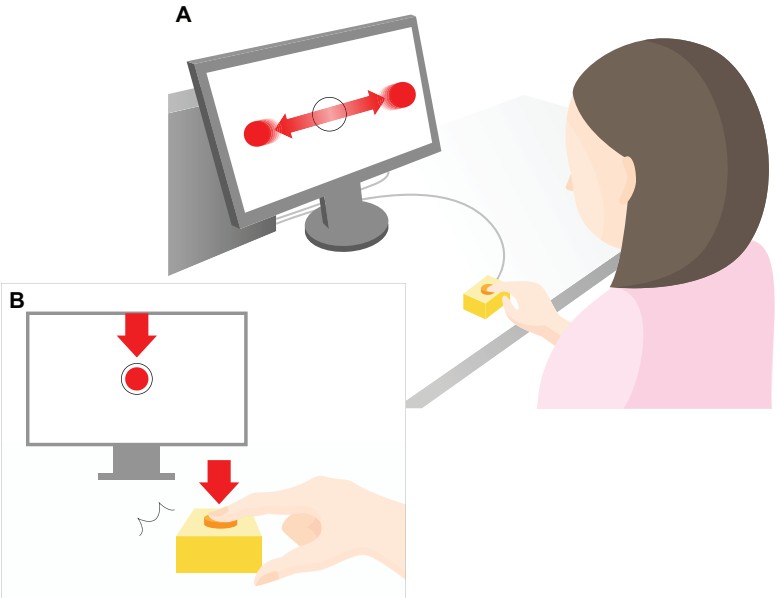

**Figure 1 Experimental setup.** A 23-inch monitor (screen size 1,098 × 630 px) was used to display the task to the participants. As shown in the top part of the figure (A), red, flat circular target with radius 20 pixels (px) repeatedly moved horizontally across the computer screen (indicated by the horizontal red arrows) at a constant speed (3,294 px/s). The target reciprocated the screen 1.5 times per second. The participants were instructed to press a key to restrict the ball to within a target circle (radius 30 px) at the center of the screen (B). The arrow indicates the moment at which the object stops within the target region in the center of the screen after pressing a key. The distance between the center of the ball and the center of the target was measured in px, and the mean value of all the trials within each set was used as the error value for that set. During the experimental task only, a "beep" was played after a randomly chosen delay of 200, 500, or 700 ms after the participant hit the key. The participants were instructed to estimate the delay duration, and the difference between the estimated and actual delay was defined as the IB value.

## Preliminary task

The participants first completed a preliminary practice exercise to familiarize themselves with the task set-up. To practice estimating the time interval after pressing the key, there was a delay of 1–1,000 ms before the tone (50 ms, 900 Hz) was played. This preliminary task was administered to the participants over the course of 18 trials. The time delay was randomly selected for each trial and each participant received feedback on the actual delay after making their response (Fig. 2A). As the preliminary task was a training task, it was excluded from subsequent analyses.

## Control task

A control task was established based on the designs used in previous studies (*Poonian & Cunnington, 2013*; *Poonian et al., 2015*) to control for individual differences in time perception. We referred to previous studies (*Berberian et al., 2012*; *Humphreys & Buehner, 2009*; *Fereday & Buehner, 2017*) to determine the appropriate number of trials required for our study. In our control task, the participants performed 18 trials where they had to estimate the time interval between two sounds. The interval between the two sounds was either 200, 500, or 700 ms, which was chosen at random for each trial. The delay at

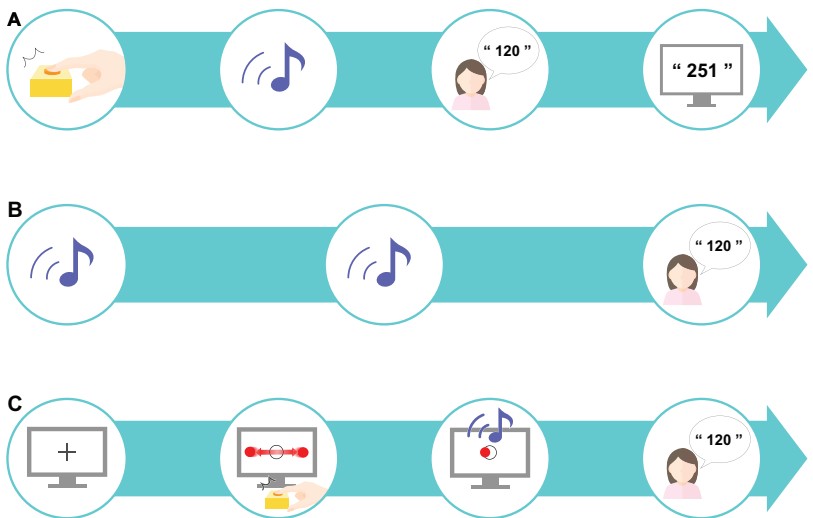

**Figure 2 Experimental procedure.** (A) A preliminary task was designed to promote an understanding of estimating the sense of time. A sound was played after a time delay (ranging from 1 to 1,000 ms) after the participant pressed the key; the participant was asked to estimate the delay interval and was then given feedback about their accuracy. (B) The control task tested the estimation of delay interval in an unrelated context to the perceptual-motor task. This task only included sounds: two sounds were presented after a randomly chosen delay of 200, 500, or 700 ms, and the participant was asked to estimate the delay interval. The mean value of the differences between the estimated and actual delay interval was set as the baseline value. (C) The experimental task was designed as a "real-life" task. First, a fixation cross was presented on the computer screen for 1 s, after which a moving horizontal ball across a target circle was displayed. When the participants pressed the key to indicate that the moving ball was on the center of the target, a sound was presented after a randomly chosen delay of 200, 500, or 700 ms. The participant was instructed to estimate the delay interval. This experimental task comprised 10 sets, each consisting of 18 trials.                                                                               

these three differing levels was presented six times each, comprising 18 trials in total. Our task had the additional goal of investigating learning: if too many trials were performed, learning would be finished within 1 set, which we assumed would not be enough to capture the temporal changes. Thus, to capture both IB and learning, we set the number to 18 trials. The difference in the actual delay duration and the estimated duration was calculated and the mean value from the 18 trials was calculated as the baseline. Here, the participants did not receive any feedback on the actual time delay (Fig. 2B).

## Experimental task

In the experimental task, the participants performed an IB task that included elements of a perceptual-motor task. The participants were first presented with a black fixation cross for 1 s, followed by a circular, flat red object that moved horizontally across the screen (1.5 times per sec) at a constant speed (3,294 px/s) traversing a target circle in the middle of the screen. The participants were instructed to press a key when the object reached the center of the target circle. The circular target had a radius of 20 px and the circle in the center of the screen had a radius of 30 px. The target stopped moving as soon as the participant pressed the key. A sound was then presented to the participant after a delay and the participant had to estimate the delay between pressing the key and hearing the sound (Fig. 2C). One set consisted of 18 trials and a total of 10 sets were

performed. In this experimental task, actual delays of 200, 500, or 700 ms were randomly presented. As with the control task, these three levels of delay were presented six times each to comprise the 18 trials in total. The difference between the actual delay duration and the estimated duration was calculated, and the mean value for each set (all 18 trials) was calculated.

The distance in pixels between the center of the moving circular object and the center of the target circle was measured and used as the behavioral index; the mean value for each set represented the error. The error value was used as an index for perceptual-motor performance. The value obtained when the actual delay duration was subtracted from the estimated delay duration represented the IB value. We made the assumption that a low IB value indicated a high SoA (IB effect).

### Data analysis

The participants were clustered based on their perceptual-motor performance characteristics using Ward's method, where the error values obtained from the first and last set, not by overall amount of error in the experimental task were used as the variables. To compare the error in each cluster, the 10 sets were divided into five blocks (blocks 1–5) containing two sets each (e.g., block 1 is the mean value of sets 1 and 2).

Perception of time varies depending on the participant, thus to eliminate any bias associated with individual sense of time, the mean value of each set from the control task was subtracted from the IB value for each set from the experimental task: *(estimated time in the experimental task—actual time in the experimental task)—(estimated time in the control task—actual time in the control task)*.

To avoid type II errors (false-positives), the 10 sets were divided into five blocks (blocks 1–5) containing two sets each. Analysis using the Shapiro–Wilk test showed that one part of these blocks was not normally distributed. Therefore, comparisons of the error and IB value in each block were performed using the Friedman test, and the Wilcoxon signed-rank test was used to correct for multiple within-group comparisons. Additionally, the Mann–Whitney $U$-test was used to correct for multiple comparisons between groups. In addition, the average values of IB and errors in all sets were calculated for each participant. Using these values, the Pearson's correlation coefficient between the IB value and the error value was calculated for cluster 1 and cluster 2. $p$-values $< 0.05$ were considered statistically significant, and the Bonferroni correction was used for multiple comparison adjustment. The R statistical software package was used for all statistical analyses (*R Core Team, 2017*).

### RESULTS

Cluster analysis using the error values obtained from the first and last sets classified the participants into two groups (Table 1): cluster 1 ($n = 19$) and cluster 2 ($n = 11$) (Fig. 3). The mean difference in error between the first and the final (10th) set of each task for participants in cluster 2 (mean $= -86.7$, SE $= 15.8$) was significantly higher than those in cluster 1 (mean $= -31.3$, SE $= 7.2$) ($t$-value $= -3.61$, d$f = 28$, ES ($r$) $= 0.56$, $p = 0.001$) (Fig. 4). This finding means that the error reduced as the task progressed, indicating that

**Table 1 Statistics of each cluster.**

|  | Cluster 1 | Cluster 2 |
|---|---|---|
| Vector of within-cluster sum of squares, one component per cluster | 54,558.370 | 25,703.370 |
| Total within-cluster sum of squares | 80,261.74 | |
| Between-cluster sum of squares | 172,674.10 | |
| Correlation coefficient | −0.107 | −0.660 |
| $p$-value | 0.663 | 0.027 |
| Degrees of freedom | 17 | 9 |
| 95% confidence interval | −0.535–0.365 | −0.903 to −0.100 |

**Note:**
Participants were classified into two groups, cluster 1 ($n$ =19) and cluster 2 ($n$ = 11), based on a cluster analysis using the error values obtained from the first and last sets.

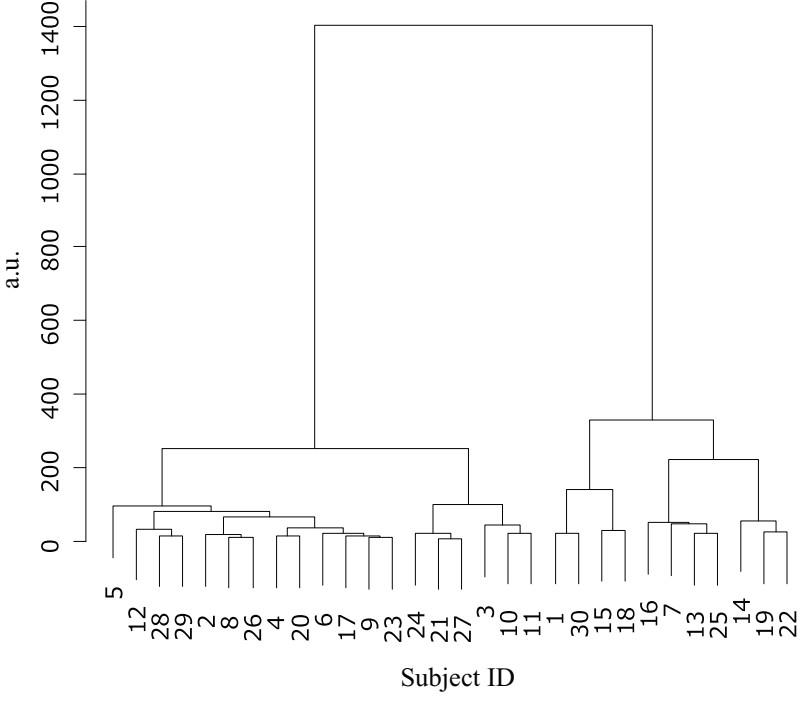

**Figure 3 Cluster analysis.** Dendrogram based on the cluster analysis generated using Ward's method, wherein the error values from the first and final sets were used as variables. The participants were subsequently divided into two clusters.

participants in cluster 2 exhibited an improvement in perceptual-motor performance during this task.

We analyzed the variation in the error across the five blocks. In the within-group comparisons, participants in cluster 1 showed a significantly lower error value in blocks 4 (rank-difference = 1.684, ES ($r$) = 0.533, 95% CI = 0.257–0.728, $p$ = 0.005) and 5 (rank-difference = 1.579, ES ($r$) = 0.499, 95% CI = 0.214–0.706, $p$ = 0.009) compared to block 1. Participants in cluster 2 showed a significantly lower error value in blocks 4 (rank-difference = 2.364, ES ($r$) = 0.747, 95% CI = 0.476–0.889, $p$ = 0.015) and 5 (rank-difference = 2.545, ES ($r$) = 0.805, 95% CI = 0.580–0.916, $p$ = 0.029) compared to block 1
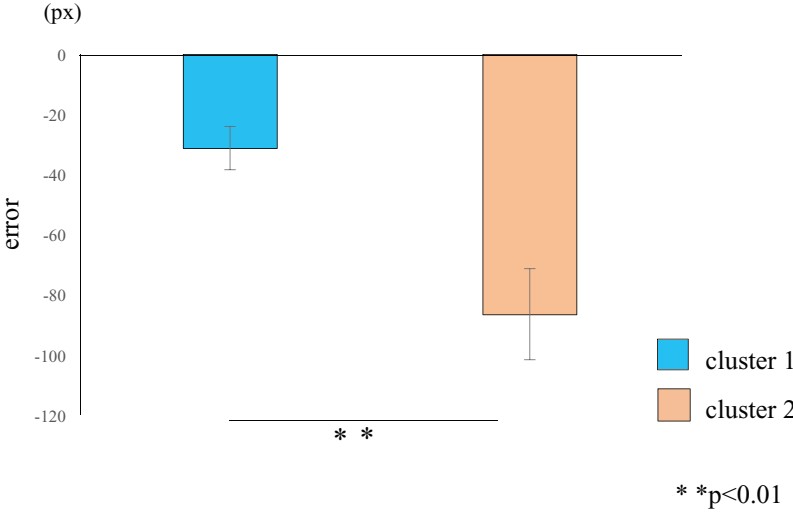

**Figure 4 Cluster-wise comparisons of the differences in error values.** The difference in the mean errors in cluster 2 (orange) was significantly higher than the difference in the mean errors in cluster 1 (blue), thus demonstrating a learning effect in participants classified into cluster 2. The data represent the means ± standard error. px, pixel.     

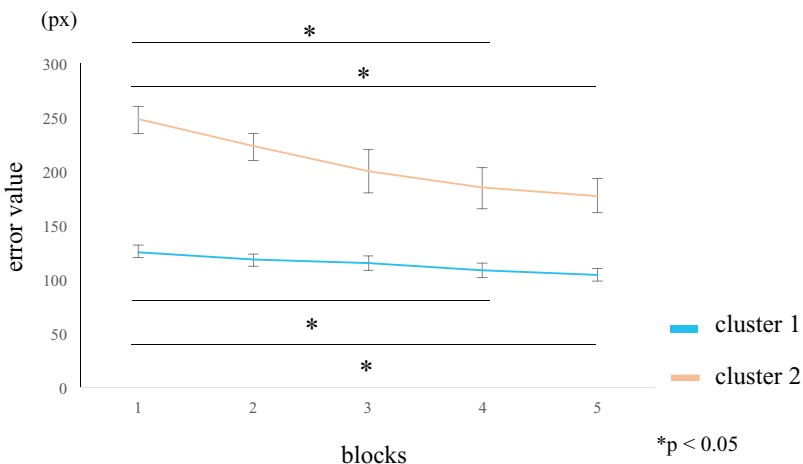

**Figure 5 Block-wise transition of error values.** Participants in cluster 1 (blue line) show low error values in block 1, and small difference in the error values among blocks. Participants in cluster 2 (orange line) show high error values in block 1 that decrease over the course of the task. A within-group comparison between clusters 1 and 2 showed a significant error reduction in blocks 4 and 5 than in block 1. However, the effect sizes for blocks 4 and 5 were larger in cluster 2 than in cluster 1 (see Table S2). The data represent the means ± standard error. px, pixel.

(Fig. 5; Table 2). The difference in the error values between the first and final set was greater in cluster 2 than in cluster 1. In the within-group comparisons, the effect sizes were larger in cluster 2 than in cluster 1 (Table 2). Therefore, the participants in cluster 2 likely exhibited a high perceptual-motor learning effect. In the between groups comparison, the error values were significantly larger in cluster 2 than in cluster 1, for all blocks (Table 3).

The IB value for the participants in cluster 2 was lower in block 5 compared to block 2 (rank-difference = 2.455, ES = 0.776, 95% CI = 0.527–0.902, $p = 0.044$) (Fig. 6; Table 4). By

**Table 2 Within-group comparisons of error.**

| Cluster 1 | | | | | Cluster 2 | | | | |
|---|---|---|---|---|---|---|---|---|---|
| Freedman test | | | | | Freedman test | | | | |
| $\chi^2$ | 14.484 | | | | $\chi^2$ | 18.909 | | | |
| d$f$ | 4 | | | | d$f$ | 4 | | | |
| $p$ | 0.006 | | | | $p$ | 0.001 | | | |
| ES ($\eta^2$) | 0.152 | | | | ES ($\eta^2$) | 0.344 | | | |
| 95% CI | 0.059–0.373 | | | | 95% CI | 0.150–0.803 | | | |
| **Multiple comparisons** | | | | | **Multiple comparisons** | | | | |
| Level (block) | Rank-difference | ES ($r$) | 95% CI | $p$-value | Level (block) | Rank-difference | ES ($r$) | 95% CI | $p$-value |
| 1–2 | 0.684 | 0.216 | −0.111–0.501 | 0.456 | 1–2 | 1.091 | 0.345 | −0.090–0.669 | 1.000 |
| 1–3 | 1.053 | 0.333 | 0.015–0.590 | 0.161 | 1–3 | 1.727 | 0.546 | 0.162–0.787 | 0.205 |
| 1–4 | 1.684 | 0.533 | 0.257–0.728 | 0.005 | 1–4 | 2.364 | 0.747 | 0.476–0.889 | 0.015 |
| 1–5 | 1.579 | 0.499 | 0.214–0.706 | 0.009 | 1–5 | 2.545 | 0.805 | 0.580–0.916 | 0.029 |
| 2–3 | 0.368 | 0.117 | −0.211–0.421 | 0.473 | 2–3 | 0.636 | 0.201 | −0.241–0.574 | 1.000 |
| 2–4 | 1.000 | 0.316 | −0.004–0.578 | 0.179 | 2–4 | 1.273 | 0.402 | −0.023–0.705 | 1.000 |
| 2–5 | 0.895 | 0.283 | −0.040–0.553 | 0.243 | 2–5 | 1.455 | 0.460 | 0.048–0.738 | 0.073 |
| 3–4 | 0.632 | 0.200 | −0.128–0.488 | 0.437 | 3–4 | 0.636 | 0.201 | −0.241–0.574 | 1.000 |
| 3–5 | 0.526 | 0.166 | −0.162–0.462 | 0.457 | 3–5 | 0.818 | 0.259 | −0.183–0.613 | 1.000 |
| 4–5 | −0.105 | −0.033 | −0.349–0.289 | 0.419 | 4–5 | 0.182 | 0.057 | −0.373–0.468 | 1.000 |

Notes:
A significantly lower error value was observed for blocks 4 and 5 than for block 1 for participants in clusters 1 and 2. Cluster 2 had higher effect sizes between block 1 and block 4 and 5.
d$f$, degrees of freedom; ES, effect size; CI, confidence interval.

**Table 3 Comparisons between groups of error.**

| Block | Rank sum | | Mann–Whitney $U$ | $Z$ | $p$-value |
|---|---|---|---|---|---|
| | Cluster 1 | Cluster 2 | | | |
| 1 | 10.00 | 25.00 | 209 | 4.497 | 0.000 |
| 2 | 10.37 | 24.36 | 202 | 4.196 | 0.000 |
| 3 | 11.16 | 23.00 | 187 | 3.551 | 0.002 |
| 4 | 11.68 | 22.09 | 177 | 3.120 | 0.018 |
| 5 | 11.42 | 22.55 | 182 | 3.335 | 0.007 |

Note:
Mann–Whitney $U$-test was used to correct for multiple comparisons between groups. Cluster 2 had a significantly larger error value for all blocks than cluster 1.

contrast, the IB value for participants in cluster 1 did not significantly differ between the blocks. These data suggest that the IB effect only increased in participants within cluster 2. In the between-groups comparison, there were no significant differences in the IB values of both clusters. The Supplemental Figure shows the values of the two clusters by actual intervals.

Although not statistically significant, we found a negative correlation trend between the average IB value and the average of error in cluster 2 only ($r = -0.53$, $p = 0.09$) (Fig. 7).

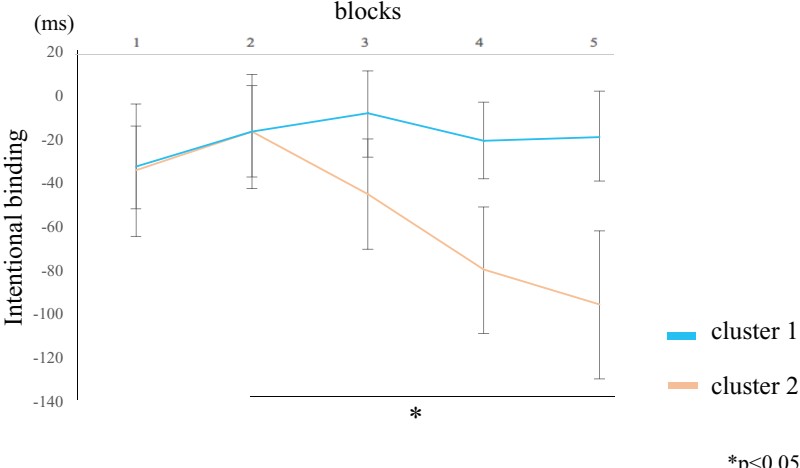

Figure 6 **Block-wise transitions in intentional binding (IB).** Within-group comparisons revealed no significant difference in the IB values between blocks for participants in cluster 1 (blue line). The IB values decreased with time for participants in cluster 2 (orange line), such that the IB value of block 5 was significantly lower than that of block 2. The data represent the means ± standard error.

Table 4 **Within-group comparisons of intentional binding.**

| Cluster 1 | | | | | Cluster 2 | | | | |
|---|---|---|---|---|---|---|---|---|---|
| **Freedman test** | | | | | **Freedman test** | | | | |
| $\chi^2$ | 5.005 | | | | $\chi^2$ | 16.873 | | | |
| d$f$ | 4 | | | | d$f$ | 4 | | | |
| $p$ | 0.287 | | | | $p$ | 0.002 | | | |
| ES ($\eta^2$) | 0.053 | | | | ES ($\eta^2$) | 0.307 | | | |
| 95% CI | 0.000–0.194 | | | | 95% CI | 0.129–0.743 | | | |
| **Multiple comparisons** | | | | | **Multiple comparisons** | | | | |
| **LEVEL (block)** | **Rank-difference** | **ES ($r$)** | **95% CI** | **$p$-value** | **Level (block)** | **Rank-difference** | **ES ($r$)** | **95% CI** | **$p$-value** |
| 1–2 | −0.947 | −0.300 | −0.565–0.022 | 0.292 | 1–2 | −0.545 | −0.172 | −0.554–0.269 | 1.000 |
| 1–3 | −1.000 | −0.316 | −0.578–0.004 | 0.256 | 1–3 | 0.545 | 0.172 | −0.269–0.554 | 1.000 |
| 1–4 | −0.658 | −0.208 | −0.495–0.120 | 0.699 | 1–4 | 1.273 | 0.402 | −0.023–0.705 | 0.806 |
| 1–5 | −0.816 | −0.258 | −0.534–0.067 | 0.447 | 1–5 | 1.909 | 0.604 | 0.244–0.817 | 0.147 |
| 2–3 | −0.053 | −0.017 | −0.335–0.305 | 0.459 | 2–3 | 1.091 | 0.345 | −0.090–0.669 | 0.806 |
| 2–4 | 0.289 | 0.092 | −0.235–0.400 | 1.000 | 2–4 | 1.818 | 0.575 | 0.202–0.802 | 0.102 |
| 2–5 | 0.132 | 0.042 | −0.282–0.357 | 0.798 | 2–5 | 2.455 | 0.776 | 0.527–0.902 | 0.044 |
| 3–4 | 0.342 | 0.108 | −0.219–0.414 | 1.000 | 3–4 | 0.727 | 0.230 | −0.212–0.594 | 1.000 |
| 3–5 | 0.184 | 0.058 | −0.266–0.371 | 1.000 | 3–5 | 1.364 | 0.431 | 0.012–0.722 | 0.206 |
| 4–5 | −0.158 | −0.050 | −0.364–0.274 | 1.000 | 4–5 | 0.636 | 0.201 | −0.241–0.574 | 1.000 |

Notes:
The intentional binding (IB) value of participants in cluster 2 was lower for block 5 than for block 2. By contrast, the IB value of participants in cluster 1 did not significantly differ among the blocks.
d$f$, degrees of freedom; ES, effect size; CI, confidence interval.

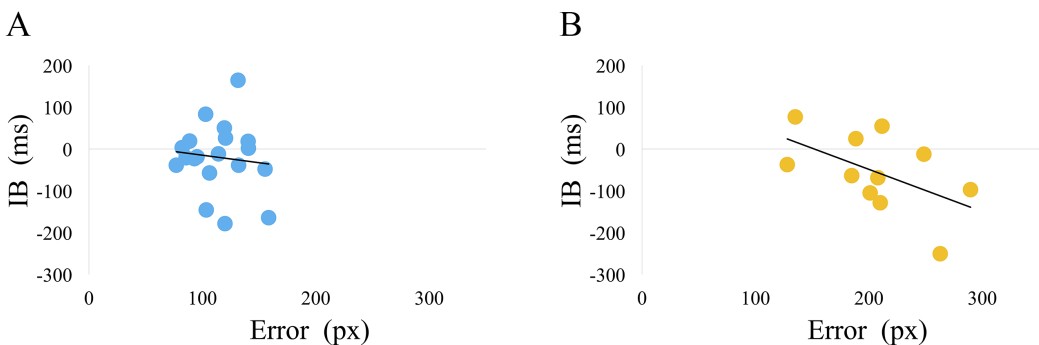

**Figure 7 The relationship between the average IB value and the error.** (A) cluster 1, (B) cluster 2. The average IB value and error in all sets was calculated for each participant. Using these average values, the Pearson's correlation coefficient between the average IB and the error was calculated for clusters 1 and 2. There was no statistically significant correlation for either clusters (cluster 1: $r = -0.10$, $p = 0.66$, cluster 2: $r = -0.53$, $p = 0.09$). Cluster 2, however, showed a trend for a negative correlation.

## DISCUSSION

The aim of this study was to elucidate how IB is modulated over time during a task involving perceptual-motor learning. Conventional IB tasks used to date do not consider the possibility that perceptual learning would change the degree of binding. Although similar mechanisms exist for a perceptual-motor learning process (where the learners modify their behavior through error correction) and for IB, these processes have typically been studied separately. In our novel approach, we developed an experimental task that is capable of detecting binding effects during perceptual-motor learning and analyzed the time-series data of the learning process. In this newly-created perceptual-motor task, the learners were instructed to observe a circular target that moved horizontally across a computer screen at a consistent speed, and stop the target (by pressing a key) when it lay within a circle in the center of the screen. The distance between the target object from the center of the circle was measured and the decrease in that distance over the course of the trial was defined as the perceptual-motor performance improvement.

Many experimental protocols use a Likert-type scale for participants to report the level of self-agency (or non-agency) they experience over a particular action or outcome. Some have proposed that IB can, however, be assessed implicitly when SoA assessment is not the primary objective. In such cases, the participant is asked to perform another type of task, and SoA is inferred from the results (*Hon, 2017*). In our study, we favored this implicit approach because the perceptual-motor and IB tasks were concurrently administered, and the learning process in the perceptual-motor task pre-supposes that the behavior is performed by the learner. Previous studies have also required a key press at a specific time including studies by *Kumar & Srinivasan (2017)* and *Vinding, Pedersen & Overgaard (2013)*, of which the latter used an interval estimation task. We developed our IB task during perceptual-motor learning, in accordance with the task set-up used by *Kumar & Srinivasan (2017)*. We consider that the learning effect of this task reflects the learning effect of the timing of movement execution based on visual information.

A major aim of our study was to observe the changes in the binding effect over time during a perceptual-motor task. Di Costa et al. (2018) investigated the binding effect of reward during performing a learning task and found that the binding effect was enhanced in the "no reward on trial" compared to the "reward on trial." They speculated that errors can guide adjustments for the next action, and found that agency increased. However, if errors increase the binding effect regardless of rewards, the binding effect should increase in both the rewarded and non-rewarded trials. The degree of error may change the binding effect; therefore, it is necessary to distinguish between those who committed an error and those who did not during the perceptual-motor task and see if the binding effect varies among them. Here, we used cluster analysis to distinguish between these groups and then investigated whether the changes in IB with time were observed when error decreased in the perceptual-motor task. As such, we differentiated participants based on individual performance abilities.

Using the error value during the perceptual-motor task, the participants were categorized into two groups: cluster 1 and cluster 2. Cluster 1 represented the low-learning effect group while cluster 2 represented a high-learning effect group. Those in cluster 2 showed signs of learning based on a pattern of error values; the error started high in the first block of the task and then decreased significantly over subsequent trials. The difficulty level of the perceptual-motor task used in the current study was thus considered optimal for participants in cluster 2, but not for those in cluster 1.

No significant differences in the binding effect were found between all blocks of both clusters 1 and 2; however, the binding effect gradually increased after block 2 in cluster 2 participants. As all participants performed the same action, the IB value was similar between clusters 1 and 2 in the early phases of the trial (i.e., blocks 1 and 2), but IB changed with time in accordance with the decrease in error value only in cluster 2. Furthermore, although there was no statistical significance, we observed a trend of correlation between the average IB value and the average error value for cluster 2 only. Because the action was the same for the participants in both clusters, it is unlikely that the predictive mechanism directly attributed to increased binding. For cluster 2 participants, it is possible that the task's difficulty level was optimal for them to make a certain amount of errors in the early phases of the trial; these errors may have acted as a penalty. These participants then elicited fewer errors as the trials were repeated, which may have acted as a reward and eventually affected binding. Therefore, of the results obtained from cluster 2 partially support the study of Di Costa et al. (2018).

Because the participants in this experiment were not explicitly provided with the performance feedback or the amount of rewards, we cannot clearly assign the extent by which the shift in error was deemed rewarding. However, the task was designed in such a way that the participants could tell the location where they had stopped the object upon pressing a key, that is, the difference in the position of the object relative to the target in the center of the screen. Thus, it could be inferred that the participant considered the shift in error as a reward. Here, we speculate that the dopaminergic reward system was activated only in participants of cluster 2 in which a time-series variation for errors was seen. This is because the participants experienced a reduction in errors via the

repeated trials (i.e., reward outcome) relative to the amount of errors originally predicted (i.e., reward prediction). As we did not measure changes in the amount of dopamine released, dopaminergic neuron activation, or the perceived levels of motivation, we can only speculate on this possible role of dopamine and motivation in the mechanisms underlying this process.

Our data suggest that the increased binding effect in cluster 2 participants was driven by a perceptual-learning process, whereby the perceptual-motor task was performed to achieve a goal as a reward. Goals and rewards are highly associated, and it has been suggested that a goal-directed action impacts agency (*Wen, Yamashita & Asama, 2015*). We speculate that greater error values during the perceptual-motor task in cluster 2 relative to cluster 1 participants led to a stronger intention to adjust the behavior to correct these errors, which in turn contributed to the differences in binding effect between the two groups. As a result, we suggest that an IB effect was observed while perceptual motor learning proceeded and the error frequency decreased; once learning ceased, the IB effect reduced.

A limitation to our study is that the timing of the key press depended on the participants, and the predicted time was not constant. As such, the predicted time may have influenced the binding effect. *Barlas & Obhi (2013)* reported that having the freedom to choose one's action increases agency. In our perceptual-motor task, the procedure allowed the participant to press the key at their preferred timing, respecting the volition of the participants. Future studies must verify the binding effect while controlling the predicted time. In so doing, setting up a novel control task for key pressing without requiring a perceptual-motor task would allow us to further explore the impact of prediction on IB during the perceptual-motor task. In addition, although this study showed the relationship between learning effect at refresh rate of 60 Hz and IB, a refresh rate >100 Hz may result in a different learning speed. Studying the relationship between these learning variables and SoA is now required. Our experimental task included elements of intention to act, action-related proprioceptive feedback, individual difference in the perceived duration between a visuo-motor event and a tone, motor performance of each trial, and improvement in motor performance. Conversely, the factors mediated by the control task were intention to act, action-related proprioceptive feedback and individual difference in the perceived duration between a visuo-motor event and a tone, but some potential variables have yet to be controlled. The fact that some potential variables are not controlled is a limitation of our study and an area for future research.

## CONCLUSIONS

We consider that we have successfully developed a novel IB task that includes a perceptual-motor task. This task allowed us to observe temporal changes in binding effects (in the context of perceptual-motor performance improvements). Cluster analysis identified two groups of participants despite the fact that we used only one task. The first cluster included those who showed a low initial incidence of error in the first block that did not markedly change after repetition. The second cluster showed a high incidence of error in the first block but exhibited a gradual decrease in error as the trials were

repeated. We hypothesized that an IB effect is not seen in participants that find the perceptual-motor task easy (cluster 1). Conversely, an IB effect is seen in participants that find the task is at an optimal difficulty level thus resulting in the optimal amount of errors (cluster 2). Overall, those who exhibited progress in learning (cluster 2) showed an increase in binding effect, while those who did not show clear evidence of learning (cluster 1) demonstrated stable binding values. With the incidence of errors acting as a penalty, a reduction in the number of errors through repeated trials acted as a reward. This reward motivated the participant to try to reduce the error further and thus likely enhanced their attention to the task. The increase in binding effect observed in cluster 2 participants may, therefore, be due to the action of the reward or the attention system. Future studies remain to definitively determine the underlying mechanism that is responsible for the increased binding effect. Future work will also investigate whether binding effects would increase in cluster 1 participants if the task was made more difficult (e.g., by increasing the speed of the horizontally moving target, randomly changing the speed of the target, or making the central circle smaller). Other studies should examine whether the increase in binding effect observed in cluster 2 would be diminished once the participants adapt to the task through repetition and subsequently produce almost no error.

### Funding

This work was partially supported by JSPS KAKENHI, Grant-in-Aid for Scientific Research on Innovative Areas "Understanding brain plasticity on body representations to promote their adaptive functions" (Grant Number 15H01671). There was no additional external funding received for this study. The funders had no role in study design, data collection and analysis, decision to publish, or preparation of the manuscript.

### Grant Disclosure

The following grant information was disclosed by the authors:
JSPS KAKENHI, Grant-in-Aid for Scientific Research on Innovative Areas "Understanding brain plasticity on body representations to promote their adaptive functions": 15H01671.

### Competing Interests

The authors declare that they have no competing interests.

### Author Contributions

- Shu Morioka conceived and designed the experiments, performed the experiments, analyzed the data, contributed reagents/materials/analysis tools, prepared figures and/or tables, authored or reviewed drafts of the paper, approved the final draft.
- Kazuki Hayashida conceived and designed the experiments, performed the experiments, analyzed the data, contributed reagents/materials/analysis tools, prepared figures and/or tables, authored or reviewed drafts of the paper, approved the final draft.

- Yuki Nishi performed the experiments, analyzed the data, contributed reagents/materials/analysis tools, prepared figures and/or tables, authored or reviewed drafts of the paper, approved the final draft.
- Sayaka Negi performed the experiments, analyzed the data, prepared figures and/or tables, authored or reviewed drafts of the paper, approved the final draft.
- Yuki Nishi analyzed the data, approved the final draft.
- Michihiro Osumi conceived and designed the experiments, analyzed the data, contributed reagents/materials/analysis tools, approved the final draft.
- Satoshi Nobusako conceived and designed the experiments, contributed reagents/materials/analysis tools, approved the final draft.

## Human Ethics

The following information was supplied relating to ethical approvals (i.e., approving body and any reference numbers):

The experimental protocol was approved by Kio University Ethics Committee (Ethical Application Ref: H28-50).

## Data Availability

All raw data are included in Supplemental File 1.

## Supplemental Information

Supplemental information for this article can be found online at http://dx.doi.org/10.7717/peerj.6066#supplemental-information.

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
