# Peer review of "Changes in intentional binding effect during a novel perceptual-motor task"

_PeerJ, doi:10.7717/peerj.6066_

## Round 0.1 · original submission · Major Revisions

Although the study is of interest, the reviewers raised many questions that need to be addressed before it can be determined whether the study meets the criteria for publication in PeerJ. In particular, the authors should clarify the rational of the study and explain what is learned by participants, justify the control task and the timing of the sound presentation, clarify the statistical analyses that were preformed, report the interaction effect and consider performing a regression analysis, and review the writing for clarity and grammatical correctness.

·

Basic reporting

The authors investigated how the intentional binding is modulated through progress of perceptual-motor learning. Instead of using self-paced action task in the original studies by Haggard, authors created a task which needs practice to do successfully. Participants were grouped depending on how they improved their performance. The question is interesting.

There are two major concerns. First, the control task was not well-designed. The authors aimed at investigating ‘sense of time not associated with IB’ in the control task. If this was true, the only difference between the main and control tasks should be the occurrence of IB. However, the control task of this study is different from the main task in visual and auditory stimuli and tactile feedback. In addition, the number of trials of the control task was only 18 though. It is hard to say what the results of the subtraction (line 192) express.

Second, in this study, the timing of the presentation of the sound was randomly delayed. I am wondering if longer delay reduces the sense of agency and vice versa.
As far as I know, previous studies have shown that the timing of a sound resulted by one’s action is perceptually bound to the action. In short, motor actions cause the intentional binding (motor actions -> IB). This does not mean that sense of agency causes the intentional binding (motor action -> SoA -> IB). Instead, shortened perceived time could also cause the sense of agency (motor action -> IB -> SoA).
I hope the authors explain that there is no need to worry the possibility that the manipulated delay of sensory feedback contaminates the sense of agency and justify the methodology of this manuscript.

Minor comments

Title page
Do the third and fifth authors have same names?

Abstract (line 33)
What change is thought to be an improvement is not clear.

Introduction line 57 and Discussion line 234
In my understanding, the underlying mechanism of temporal binding is still to be investigated. Please refer the papers based on which the authors have understood that the perceptual-motor learning and SoA generation have similar mechanisms.

Devices and software
Please write the refresh rate of the display.

Experimental task
Please write the size of the stimuli in visual angle (deg).

Figure 1
I was not able to understand the figure. Please clarify what the arrows indicate.

Experimental design

Please see 1.

Validity of the findings

no comment

Reviewer 2 ·

Basic reporting

The authors reviewed literature well and reported interesting findings supported by their data. I hope my comments below will be helpful.

Was the manuscript proofread by native, professional English speaker? I do not think English in the manuscript is intelligible and professional. Example follows:
Line 25: After what? Though I know it means 200-700 ms after the keypress, it should be revised.
L28: What do “sets” mean?
L33: should be revised: “improvement” not “changes (improvement)”.

L57: I did not understand how learning and agency share similar mechanisms. Do you mean reward and motivation? If so, you may need to mention here the internal-prediction and/or comparator account for the generation of agency.

L76: This sentence is misleading because IB is an implicit measure of agency but does not directly reflect explicit awareness of agency, in nature (though IB and explicit measure “can be” related each other in some situations).

L83: Please describe and cite precisely as much as possible. See below:
Outcome binding–Postdictive process: Moore, J. W., Lagnado, D., Deal, D. C., & Haggard, P. (2009). Feelings of control: contingency determines experience of action. Cognition, 110(2), 279-283. doi:10.1016/j.cognition.2008.11.006
Action binding–Predictive process: Moore, J. W., & Haggard, P. (2008). Awareness of action: Inference and prediction. Consciousness and Cognition, 17(1), 136-144. doi:10.1016/j.concog.2006.12.004
Combination of the two: Moore, J. W., Wegner, D. M., & Haggard, P. (2009). Modulating the sense of agency with external cues. Consciousness and Cognition, 18(4), 1056-1064. doi:10.1016/j.concog.2009.05.004

L86: I guess the emotional aspect (ie. Yoshie, Takahata) should be discussed in the following paragraphs.

L139: As detailed information is missing, I think the description in Fig 1 caption should be moved to the main text. Viewing distance should be reported.

L146: Readers may misunderstand as a “computational” model.

L147: Moore & Obhi 2012 is just a review paper. I guess interval estimation method for the first time was employed in the following study: Engbert, K., Wohlschlager, A., Thomas, R., & Haggard, P. (2007). Agency, subjective time, and other minds. Journal of Experimental Psychology: Human Perception and Performance, 33(6), 1261-1268. doi:10.1037/0096-1523.33.6.1261

L147: What does “detect causal relationship” mean? Simply, your task includes keypress action and its auditory consequences, and assesses motor performance and IB.

L163: Number of trials for each delay condition should be reported (although we can easily assume 6 trials each).

L184: How did you analyzed the 3 delays (200/500/700 ms)? As a within-factor? Or collapsed them?

L187: A plausible reason for dividing 10 sets into 5 blocks should be mentioned, as the clusters were determined based on the 1st and 10th sets. Perhaps, you divided to avoid underpower (type 2 error) due to multiple comparison between 10 sets?

L204: Suppl tables were cited several times, and I think these data are important for understanding the results well. So, if available, the authors should include these tables in the main text.

L289: This sentence seems to be too conclusive: which “may have” acted as?

L308: Demanet 2013 and Barlas 2013 showed effects of physical effort and action options on agency but not effect of attention. Please correct them and/or add appropriate citation if needed.

L316: It should be temporal judgment task, not agency judgment task.

L331: did not “show” major?

L350: What do the “predicted time” and “simulation time” mean? Please specify.

Overall: the authors use the term “post-error” as if it is adverb (e.g. occurs post-error; effect post-error for). But is it grammatically correct?

Suppl Data: Non-English words were found in the “cluster” sheet. Use English only.

Suppl Video: English subtitles may be required as she said something in non-English.

Experimental design

The authors stated that the present task is novel. But, as discussed in lines 244-258, the task is indeed partially based on the previous ones. Although novelty itself does not matter, I would recommend describing how novel the task is and how it is (un)related to the previous ones in the Introduction or Method section rather than Discussion.

This task is designed to induce perceptual-motor learning. However, it is unclear what was learned. The authors have assumed that similar sensorimotor mechanisms (i.e., internal prediction, post-hoc matching, and error correction) underlie the perceptual-motor learning and generation of agency. Thus, ideally, the present task should induce sensorimotor-like learning. In traditional perceptual-motor learning paradigms, sensorimotor coordination and motor control are learned. For example, correspondence between arm movement and its “biased” visual feedback is learned due to updates of internal model, and consequently individuals can smoothly control their arm and the feedback (e.g. rotated mouse paradigm: Imamizu et al. 2000). On the other hand, in the present task, participants viewed a moving object, expected when it comes to the center of screen, and executed one keypress movement at the desired timing. I am concerned that this task indeed induces a learning of computation and estimation of the temporal information, like a time-to-contact estimation (Tresilian, 1995). To enhance your findings, I would recommend clarifying the underlying nature of your task.

Clusters 1 and 2 were defined by the differentials between first and last sets, not by overall amount of error. Am I correct? Based on the Results section and Fig 5, both interpretation may be possible. Please clarify.

Validity of the findings

The discussion on reward is interesting but speculative. Whether amount of error is rewarding depends on whether participants know the amount of their own error. Although I understand there were no feedback about the performance, how and to what extent did the participants know the amount of error and its variation? And, to what extent did the participants feel their (varying) error as rewarding. These should be potential future directions.

L262: In my understanding, Di Costa et al. (2017) did not use a between-design (there were no groups). Please check. If so, the following discussion at L263-68 would be subject to amendment.

L284: Post-error boost is found mainly for the action binding (Di Costa et al. 2017). Action binding relies more on predictive mechanism, according to Moore & Haggard (2008). Thus, your assumption “post-error boost as a postdictive mechanism” might not be valid.

·

Basic reporting

The study investigates how intentional binding (compression of perceived duration between voluntary action and its outcome) is influenced by learning of a perceptual-motor task prior to the intentional binding task. This has been achieved by using a novel paradigm that integrates the tasks of perceptual-motor learning and intentional binding along with each other. Results show that intentional binding tends to increase over time as participant’s performance improves in perceptual-motor task. The task is novel and interesting and the results have a implication for understanding intentional binding in context to other tasks (something that majority of the IB studies ignore by studying IB in isolation), and is something that I take great interest in. Authors have not a great job in designing, conducting the study and explaining the results.
However, there are some portions with respect to methodology that need more explanation. It is my strong belief that answering these points will greatly help in improving the shape of the manuscript. I have given my comments in chronological order followed by general comments for each section.
1. Basic Reporting
a. Line 18-19: “Thus far,….investigated”. The line is not clear and doesn’t correctly reflect the experiment that the authors conducted.
b. Line 33-34: ”….IB is elevated when strong perceptual-motor learning occurs”, the word strong is not needed. In my opinion, it would be sufficient to say that IB is elevated when perceptual-motor learning occurs.
c. Line 78-81: “However, …..unclear” the sentence is unnecessarily long and complicated, I would suggest that the sentence should be rephrased to make it clearer.
d. Line 100-103: The authors talk about Di Costa et al. (2017) and rightly point that although the study shows post error boost in sense of agency they do not show how binding changes across time. However, the statement “it was also unproven whether the occurrence of errors asa postdictive mechanism enhances the IB effect” might not be the right conclusion based on the study. Results from the study clearly show that the same trial outcome does not influence the IB value. The authors should revisit the results and clearly state the original results. Authors have also missed a very important result from the study, the fact that the Di Costa et al. (2017) looked at the relationship between learning rate and IB, showing that the post error boost in IB is not a result of error in previous trial but rather a result of learning from previous trial. In my opinion authors should include the result in their manuscript as it is very closely linked to what the authors aim at investigating.
e. Line 104- Line 112: The section on dopaminergic neurons and the idea that task difficulty is linked to reinforcemnt learning looks a bit out of place and the role that the section is playing in the introduction is not clear. I would suggest either moving the section to the discussion, or making it short and integrating it with the earlier paragraph. The lines do not provide a strong enough rational to form the basis of hypothesis introduced in the next paragraph.
f. Research question needs to be specified clearly and strong rational should be provided for the research question
g. There are few grammatical errors in the manuscripts as well as few typographical erros ( line 268: from instead of form). Also, some sentences are unncessarily complex. Proof reading the document will help remove such errors in the mansucript.

Experimental design

2. Experimental Design
a. The experimental design is really interesting and definitely provides a novel approach to look the relationship between perceptual motor-task and intentional binding.
b. Line 116-117: Authors hypothesize that post-error increase in binding effects will not be present for group with low level of difficulty. However, based on the rationale authors provide for the hypothesize, same hypothesis should also be given for high levels of difficulty (or have authors decided to include only participants below a certain error thereshold)
c. Line 120: Chronoligcally, aim should preced the hypothesis. Authors may need to rewrite the introduction section in order to make these changes.
d. Line 125: “We then aimed….” The line reads as if the aim was decided in a post-hoc fashion after the initial clusters were identified. I would suggest that instead of calling it an aim of the study it can be rephrased to ….” We then investigate whether…..”. Also, it might help to keep a distinction between thing that were planned prior to study (the aim, hypothesis) and the things that were obtained post-hoc, the clusters, results. However, this is not a comment against the manuscript, how the authors choose to present the results depend on the style they are following.
e. Line 153: In the preliminary task the SOA between key press and presentation of tone is randomly selected between 1-1000 ms. However, because true randomization is being done there is a chance that participants get familiarized only with very short or very long SOA’s. This might influence how well they are able to perform the interval estimation task at later point of time.
f. Line 162: “The mean value obtained from 18 values was used as baseline”. Does the mean value here refer to the estimate given by participants or to the mean of the error between actual and reported SOA.
g. Line 165: The control task that the authros have used might not be a able to provide a pro[per baseline. It would have been great if a second control condition could be included that might measure how participants are estimating duration between keypress and outcome when they don’t perform any percpetual motor task. Only then one can talk about the effect that performing a perceptual motor task might have on estimates of intentional binding. (For example, perofrming a perceptual motor task prior to IB task might result in depletion of attentional rsources required to estimate interval in proper fashion, this would mean that results on IB do nto reflect SoA, they are rather reflecting some other cognitive process).
h. Line 185: More details should be given about the clustering procedures in the data analysis section. From text it seems that the difference in error value between first and last set was used for clustering, but I am not sure.
i. Line 188: What is the rationale of clubbing two sets together to have 5 blocks. How does decreasing the number of blocks helpful in the data analysis. Will results be different if we take 10 sets for analysis instead of 5 blocks. Why 5 blocks can we just have 2 or 3 blocks.
j. Line 192-193: Mean of time estimates from control block were taken separately for the three SOAs or results for the three SOAs was combined together.
k. Line 208-211: Authors say that difference between first and last block was significantly greater for cluster 2 compared to cluster 1 and go on to conclude that cluster 2 showed motor improvement as task progressed. However the conclusion doesn’t follow from the analysis. The t-test in line 209 suggests that learning is more in cluster 2 compared to cluster 1. In fact, supplementary table 2 shows that difference between 1st and 5th block is significant in both cluster1 and cluster 2.
l. Line 218-219. No statistical test has been conducted to support the conclusion that differnce between 1st and last block is greater in cluster 2 than cluster 1. Such a statement can only be made after analysing the interaction between cluster and block, which authors don’t look at.
m. Instead of performing cluster analysis and logistic regression would have provided more insight into the exact nature of the relationship between error and intentional binding in that trial.
n. The way data has been analyzed, it is averageing IB values as well as error values across trials within a block. Thereby not taking advantage of the time-series data available to the authors. A trial by trial analysis would have been more interesting. Also, based on current analysis it is impossible to say whether the increase in intentionalbinding is postdictive effect of error or post error boost in intentional binding (some thing that authors suggest Di Costa et al., 2017) dosent.

Validity of the findings

3. Validity of the design
a. Line 250-254: Authors talk about why libets clock cannot be used by taking a counterfactual situation stating problems with the liebts clock task. The paragraph needs to rephrased and made less complex.
b. Line 269: Authrs introudce a new classification Unsuitable and sutiable, I would suggest that authors should stick to a single convention through out the paper.
c. Line 271, 283: You have not investigated post-error boost, please modify the line.
d. Line 285-289: As post error boost is not explicitly investigated, the lines are speculations and should be menitoned as such.
e. Line 290-299: Dopaminergic explanaiton of what is happening is important but should be preceeded by discussion of behavioural studies investigating percpetual-motor learning. A complete explanation at Dopaminergic level dosen’t help as authros are not looking at that level of cognitive processes.
f. Authors should also discuss the implication of these results for mechanisms of SoA / Intentional binding.
g. A greater discussion linking other studies with control, associations, eprcpetual-motor learning and inentional bindnig with current manuscript is needed.

Additional comments

4. General Comments
a. 57-58: The authors suggest that the mechanism underlying both SoA and perceptual learning is similar. The argument in the paragraph suggests that both sense of agency and perceptual-motor learning involve actions, outcomes and are modulated by motivation. It would be wrong to say that both have similar mechanisms.
b. 63-64: it is not clear how temporal compression of interval can help individual determiner their sense of agency. In my understanding there is a correlation between IB and explicit SoA (that too not always!). There is no implication of causality in the original paper by Haggard.
c. Line 86: Authors attribute the results in Yoshie & Haggard, 2013 to rewards and penalties, the original paper attributes the effect to emotional valence and explicitly say that their results cannot be due to any kind of gain that subjects attach the outcome.

---

## Round 0.2 · Minor Revisions

While the revised manuscript has improved considerably, the reviewers have raised additional concerns that need to be addressed before the manuscript can be accepted for publication. In particular, the authors should clarify whether the experimental paradigm that is used is able to isolate improvements in motor performance or whether there are potential covariates. If so, these should be clearly acknowledged as limitations. In addition, the authors should clarify some of the reported statistics.

·

Basic reporting

1-1
Grammar of this manuscript became good now. However, I still felt difficulty in understanding what authors wanted to say very often. Some sentences did not even make sense.

For example, Line 18 'How IB may change over the course of a perceptual-motor task, however, has not been explicitly investigated' could be read as if the authors measured how the degrees of IB changes in the time course of each trial. This sentence is grammatically correct but it would be appropriate to say 'perceptual-motor learning' (instead of 'perceptual-motor task') in this context.

1-2 [Line 51-52]
The sentence is not precise.

First, we feel SoA even when we make an action spontaneously without any explicit goals or purposes. IB has been found in the experiments where participants were required to make actions spontaneously without any goals.

Second, motor control is the process to reduce the differences among them. SoA would come up when the differences among them were small.

1-3 [Line 71-73]
What Vinding and Pedersen showed was that when the intention to act is formed in advance, it entails a stronger SoA than when the intention is immediately followed by the action. Vinding and Pedersen did not mention that there was difference in the goal between two types of intention in their paradigm.

1-4 [Line 76-78]
The people who have read Haggard & Clark (2002) will guess what the authors wanted to say by these two sentences. But many readers would not understand them.

1-5 [Line 241-242]
This sentence does not make sense.
Do you mean that the conventional tasks do not consider the possibility that perceptual-motor learning would change the degree of binding?

1-6
Please check the submission guideline. You usually do not use 'et al.' to refer two-author studies.

1-7 [Line 263-268]
It was hardly possible for me to guess what the authors wanted to say.

1-8
The discussion about the reward system was a little bit speculative.

Reduction of error (i.e., increase of reward) increased SoA in the cluster 2 participants. How is the result of this study consistent with the previous study by Di Costa et al.?

1-9 [Line 340]
To conclude that the postdictive mechanism worked in the IB task, the authors need to test if the IB decreases when the feedback of the perceptual-motor task was given after the estimation of the delay of the tone.

Experimental design

2-1
The refresh rate of this study (60Hz) was very low to ask participants to make a quick motion watching a stimulus moving at high speed.
Researchers studying time perception have tried to increase the refresh rate of monitors as high as possible (>100Hz).

2-2
Time perception measured by traditional action tasks is supposed to contain the effects of
1. intention to act,
2. action-related proprioceptive feedback and
3. individual difference in the perceived timing of a tone.

Haggard measured the effect #2 by applying TMS to the motor area of participants.
Haggard and other researchers have controlled the effect #3 by passive listening conditions where participants reported timing of a tone without action execution.

In the present study, the experimental task would include the effects of
1. intention to act,
2. action-related proprioceptive feedback,
3. individual difference in the perceived duration between a visuo-motor event and a tone,
4. motor performance of each trial,
and
5. improvement in motor performance.

It is unclear which factors were controlled by the control task where the perceived duration between two tones was measured.

I suppose the aim of the study is to measure the effect #5. The effect #1-4 could be controlled by taking the performance in the first block as a baseline.

Validity of the findings

Please plot the error of the perceptual-motor task as a function of the perceived duration (perceived duration - actual duration).

Additional comments

The authors could discuss the possibility that the sense of agency plays a role in motor learning.

Reviewer 2 ·

Basic reporting

I appreciate the efforts that the authors made. The quality of the manuscript has been improved. I would like to provide some more minor comments.

L221-236:
Please specify what ES (effect size?) indicates. Cohen's d or r?

Table 1:
For cluster 1, p-value seems too high. Is this typo?

Tables 2 and 4:
For 95%CIs, zero in the ones place should not be omitted (just for consistency).

All Tables:
I recommend making the numbers of decimal places consistent. For example in Table 1, -0.6601593 -> -0.660.

Experimental design

No comment.

Validity of the findings

No comment.

---

## Round 0.3 · Minor Revisions

While the revisions have improved the manuscript, the reviewer made additional minor comments that need to be addressed before the manuscript can be accepted for publication.

·

Basic reporting

Each paragraph of the introduction part became clear and precise now.

Comment 1-1
Line 80 'Therefore, ...'
The idea described here was initially followed a sentence that Vinding et al showed that the size of IB is dependent on the purpose of the action. Now the sentence was revised. The authors need to correct the sentence following the revised sentence.

Comment 1-2
Line 82
This paragraph could be moved to line 69 (after the first sentence of the paragraph). This change will make it easier for readers to follow the flow of the logic.

Comment 1-3
Line 93
I should have commented about this point earlier but are the contributions of predictive process and of volition/planning same?

Comment 1-4
Line 130
Does authors hypothesize that the IB disappears after the participant completes motor learning process though many IB studies have shown the presence of IB in simple tasks which do not require motor learning?

Experimental design

I should have suggested this point earlier but the error values (absolute distance values without sign) do not have the information how the participants tended to press the button earlier or later than the correct timing. The tendency would reflect time perception of the participants. It would be great if the authors could plot relative error values of a few participants from the two clusters.

Validity of the findings

Comment 3-1
The authors plotted the IB values as a function of the actual interval of the stimulus to respond to my suggestion. What I wanted to suggest the authors was to plot the error values of the hand action task as a function of IB (a scatter plot with x (or y): perceived duration - actual duration [ms] and y (or x): error value [px]). If the error in the action was critical to the size of IB, the error values (y) and the IB values (x) should correlate to each other.

I would also suggest the authors to plot the IB values as a function of the change in error values of each trial compared to the former trial. Correlation between them will support the discussion that reduction in error facilitated the IB.

Comment 3-2
Line 325
This sentence could be read as if Hon et al. (2013) showed that the sense of agency is reduced when the participants were required to focus on the task very well.

Hon et al. (2013) reported that the sense of agency was low in the high cognitive load condition. Their results suggest that the sense of agency is reduced when the attention was taken away from the action and the result of the action.

Additional comments

The authors often use the word 'optimal'. I guess the authors supposed that the sense of agency could be drawn as a U shape model where too easy or too demanding second task (i.e., too low or too high requirement of attention to the second task) reduces the sense of agency. Is this understanding correct?

Line 282
This sentence sounds as if the authors did not design the task optimally to each participant.

Line 333
It was not clear what the authors thought to be 'optimal'.

---

## Round 0.4 · accepted · Accept

The authors have adequately addressed the final reviewer comments.

#